# Production and Evaluation of Chicken Egg Yolk Immunoglobulin (IgY) against Human and Simian Rotaviruses

**DOI:** 10.3390/v14091995

**Published:** 2022-09-09

**Authors:** Gentil Arthur Bentes, Natália Maria Lanzarini, Juliana Rodrigues Guimarães, Marcos Bryan Heinemann, Eduardo de Mello Volotão, Alexandre dos Santos da Silva, Luiz Guilherme Dias Heneine, Jaqueline Mendes de Oliveira, Marcelo Alves Pinto

**Affiliations:** 1Laboratório de Desenvolvimento Tecnológico em Virologia, Instituto Oswaldo Cruz, Fiocruz, Rio de Janeiro 21040-360, Brazil; 2Laboratório de Virologia Comparada e Ambiental, Instituto Oswaldo Cruz, Fiocruz, Rio de Janeiro 21040-360, Brazil; 3Departamento de Medicina Veterinária Preventiva e Saúde Animal, Faculdade de Medicina Veterinária e Zootecnia, Universidade de São Paulo, São Paulo 05508-270, Brazil; 4Laboratório de Imunologia Aplicada, Fundação Estadual Ezequiel Dias, Belo Horizonte 30510-010, Brazil

**Keywords:** immunoglobulin Y, group A rotavirus, latex bead agglutination

## Abstract

Producing specific antibodies in chickens is an attractive approach for diagnosis or therapeutic applications. Besides the high immunoglobulin Y (IgY) yield transferred to the egg yolk and its suitability for large-scale production, such an approach is more bioethical for animal maintenance. The IgY technology offers new possibilities for application in human and veterinary diagnostics and therapeutics, including strategies for treating severe intestinal diseases in children, particularly in emerging countries. Herein, we describe the production and purification of polyclonal antibodies against rotavirus group A (RVA) in immunised hens aiming at its application in prophylaxis and treatment of rotavirus-induced diarrhoea. For this purpose, we inoculated Rhodia laying chickens (*Gallus gallus domesticus*) with two or three doses of RVA combined with adjuvants or only adjuvants (control group). As the egg-laying period began, the yolk protein purification processes yielded a high concentration of specific IgY, the highest titre resulting from the group of hens that received three doses of the immunogen. The purified IgY blocked the functional activity of RVA in MA-104 cells, thus confirming the neutralisation ability. Therefore, anti-RVA IgY could be a promising candidate for pre- and post-exposure prevention or treatment of rotavirus-induced diarrhoea.

## 1. Introduction

Egg yolk immunoglobulin Y (IgY) is the most predominant antibody in birds, reptiles, and amphibians, being transferred from blood to egg yolk at high concentrations (50–100 mg/egg) [1]. Based on this property, the production of IgY in immunised hens can be a straightforward, fast, and cheap strategy for obtaining specific antibodies for diagnosis or therapeutic approaches. After immunisation, large amounts of IgY can be purified directly from a chicken egg yolk, thus avoiding invasive and painful procedures [2]. IgY presents many advantages over immunoglobulin G (IgG), such as lack of interaction with mammalian immune components, and does not interact with rheumatoid factor, human anti-mouse IgG antibodies (HAMA), complement system components, or mammalian Fc receptors. Altogether, such characteristics are worthwhile as an alternative diagnostic tool for obtaining specific antibodies [2,3].

Rotavirus (RV) is a leading causative agent of severe diarrhoea among children under five years of age worldwide, with up to 200,000 deaths a year attributed to RV infection in this age group [4]. Despite the availability of vaccines, group A rotavirus (RVA) accounts for over 90% of gastroenteritis cases [5].

RV viral particles are approximately 100 nanometres (nm) in diameter, display icosahedral symmetry, and have no envelope. RV genome comprises 11 dsRNA segments, coding for structural (VP1–VP4, VP6, and VP7) and non-structural proteins (NSP1–NP6). The complex VP2 anchored with VP1/VP3 makes up the viral core; VP6 originates the intermediate capsid, whereas VP4 and VP7 form the outer one [6]. The rotavirus virion is a triple-layered particle (TLP) which resembles wheels (rota, in Latin) by electron microscopy. The TLP is the infectious form of the virus that attaches to and enters host cells [7].

RV transmission occurs via the faecal–oral route mainly through direct or indirect contact with contaminated food, water, and fomites [8,9,10]. RV infections lead to subclinical to mild clinical manifestations, including watery diarrhoea of limited extension, or severe diarrhoea with vomiting and fever that can cause dehydration along with shock, electrolyte imbalance, and death [5]. Shedding of a large amount of virus (10^10^–10^11^ particles per gram of faeces) [11] turns RV detection easy by using a variety of immunological methods, such as enzyme-linked immunosorbent assay (ELISA), immunochromatography, and latex agglutination testing [12,13]. Herein, we describe the production and purification of specific polyclonal anti-rotavirus immunoglobulin Y in immunised hens aiming at its application in prophylaxis and treatment of rotavirus-induced diarrhoea.

## 2. Material and Methods 

### 2.1. Animals

A total of nine 38-day-old Rhodia laying chickens (*Gallus gallus domesticus*), weighing between 0.225 and 0.387 kg, were distributed in three groups of three birds. Each group of chickens received a different immunisation protocol. Hens were euthanized by total exsanguination under deep anaesthesia at 33 weeks old, 13 weeks after the beginning of the egg-laying period. We applied the animal maintenance procedures previously described by our research group using hens immunised against hepatitis A virus (HAV) [14]. The Ethics Commission on Animal Care (CETA-UNIFESO protocol no. 0331/11) reviewed and approved the experimental protocol.

### 2.2. Inoculum and Immunisation

The two immunogenic inoculums consisted of 50 µL of each: human (Wa–2.2 × 10^5^ copies) and simian (SA11–2.0 × 10^5^ copies) rotavirus, combined with 80 µL of the incomplete Freund′s adjuvant (Sigma–Aldrich, St. Louis, MO, USA) and 20 µL of CpG-ODN (1 mg/mL–5′-TCG TCG TTT GTC GTT TTG TCG TT-3′, BioCorp Inc., Montreal, Canada). The non-immunogenic inoculum consisted of 80 µL of incomplete Freund′s adjuvant and 20 µL of CpG-ODN (1 mg/mL). The human (RVA/human-tc/USA/Wa/1974/G1P [8]) and the simian (RVA/Simian-tc/ZAF/SA11/1958/G3P [2]) rotavirus used as inoculum were produced in cell culture using the African green monkey kidney lineage (MA-104-ATCC^®^ CRL-2378) and purified with caesium chloride gradient by ultracentrifugation at 240,000× *g*, as described previously [15]. The final concentrations of Wa RVA and SA11 RVA solutions were: 2.2 × 10^10^ and 2.0 × 10^8^ RNA copies/mL.

We applied three immunisation protocols: Group I received three doses of immunogenic inoculum; Group II received one dose of non-immunogenic inoculum as immune priming followed by two doses of immunogenic inoculum; and Group III received three doses of non-immunogenic inoculum (negative control). The inoculum was administered intramuscularly into the pectoral muscle. Animals received the first, second, and third immunisations at 38, 74, and 103 days old, respectively. At 138 days of the birds′ lives, egg-laying began, 34 days after the third immunisation. Eggs were collected daily for 13 weeks and stored at 4 °C. After this period, birds were euthanized (229 days old) by cardiac puncture and total exsanguination under deep anaesthesia with 40 mg/kg of thiopental sodium intravenously administered. The liver, bursa of Fabricius, kidney, spleen, bone marrow, and ileus samples were fixed in 10% buffered formalin and stored in paraffin blocks for further histological analysis (haematoxylin and eosin staining) by optical microscopy.

### 2.3. Egg Yolk Immunoglobulin (IgY) Purification and Characterisation

The egg yolks were separated from the whites and grouped into pools per week, each pool corresponding to yolks sampled from the three groups of hens. The separation of IgY from the egg yolk occurred by precipitation with polyethylene glycol 6000. After four stages of precipitation/centrifugation, the final protein precipitates were suspended in phosphate-buffered saline (PBS). Total proteins were quantified by the Lowry method [16] and IgY characterisation was based on their migration profile in sodium dodecyl sulfate-polyacrylamide gel electrophoresis (SDS-PAGE), using the Bio-Rad^®^ Mini-gel system under denaturing conditions. Purified protein pools were diluted at 1:100 in buffer/β-mercaptoethanol and denatured for 5 min at 95 °C. Next, samples were loaded into the 12% Bis-Tris gel at 120 volts for 90 min. After the gel staining with Coomassie Blue 0.1%, the IgY electrophoretic profile was identified according to the molecular weight standard BenchMark™ Protein Ladder (Invitrogen™, Carlsbad, CA, USA).

The specificity of IgY to human (Wa) and simian (SA11) rotavirus was confirmed by Western blotting and cell culture neutralisation assay. Human and simian rotavirus polypeptides were separated by SDS-PAGE as described above, and then electro-transferred onto a nitrocellulose membrane in a transfer buffer containing 20% methanol for two hours at 100 volts. Molecular weight standard Kaleidoscope™ Pre-stained Standards (Bio-Rad Laboratories, Hercules, CA, USA) were used to identify the viral proteins’ migration profile. After overnight blocking, incubation with 5% non-fat milk occurred in PBS-Tween 0.05% at room temperature under agitation, and the membranes were washed three times in PBS-T 0.05%. The membrane was cut into three strips, each containing both Wa and SA11 rotaviruses bands. The strips were incubated with the IgY purified from each group (I-III), and diluted to 0.5 mg/mL with 5% powder milk in PBS-T 0.05% for two hours at 37 °C under agitation. Next, the membranes were washed as described above. Bound IgY antibodies were detected using rabbit anti-IgY IgG conjugated with horseradish peroxidase (Sigma-Aldrich, San Luis, MO, USA) 1:2000 diluted with 5% powder milk in PBS-T 0.05% at 37 °C for one hour and thirty minutes. Afterwards, the membranes were washed as described above, and rotavirus protein bands were visualized using 0.025 g of 3,3′-diaminobenzidine tetra-hydrochloride (DAB), 100 µL of hydrogen peroxide (H_2_O_2_), and 500 µL of CoCl_2_ in 50 mL of PBS. The reaction was stopped with distilled water [17].

An in vitro neutralisation assay was performed as previously described [15]. RV was activated within 30 min at 37 °C. Serial dilutions of the anti-rotavirus IgY (0.5–0.75–1.0–1.5–2.0–2.5–3.0–3.5–4.0 mg/mL) were incubated with 3.1 × 10^7^ FFU/mL of rotavirus at 37 °C for one hour and thirty minutes. This mixture was added to the MA-104 cell monolayer and incubated at 37 °C for one hour. A rotavirus replication control was also assayed. After incubation, the cells were washed twice with PBS and kept for 24 h at 37 °C with Eagle Hanks Medium. The non-neutralised virus particles were detected by a quantitative real-time PCR technique (qRT-PCR) [15,18].

### 2.4. Conjugation of IgY to Carboxyl-Latex Beads

The anti-rotavirus IgY was covalently bound to carboxyl-latex beads (CLB) (Sigma–Aldrich, St Louis, MO, USA) activated with 1-(3-dimethylaminopropyl)-3-ethylcarbodiimide hydrochloride (EDC). First, 4.1 g of 3 µM carboxyl-latex was diluted in 1.0 mL of PBS 1X (pH 7.4). This solution was diluted in 10.0 mL of PBS, homogenised, and centrifuged at 10,000× *g* for 15 min at 4 °C, and the supernatant was slowly discarded. This step was repeated, beads were resuspended in 5.0 mL of PBS 1X. A 2 mM EDC solution 2.0 mL (50 mg/mL) was added dropwise slowly and incubated for 20 min under constant stirring. Subsequently, beads were washed with PBS 1X and centrifuged at 10,000× *g* for 5 min at 4 °C. Afterwards, 10 mg of purified anti-RVA IgY was added to 1.0 mL of carboxyl-latex, and the suspension was incubated for 3 h under constant shaking at room temperature. Finally, the CLB-bound antibody was washed with PBS 1X and centrifuged twice at 10,000× *g* for 15 min at 4 °C. A total of 10 mL of 1M ethanolamine (pH 8.0) was added and incubated overnight for blocking. Three washes were performed with PBS 1X at 10,000× *g* for 15 min at 4 °C [19]. 

The CLB-bound anti-RVA IgY conjugate was incubated with purified RVA [15]; agglutination was observed 30 min after incubation. In addition, the CLB-bound anti-RVA IgY was evaluated against a panel of RVA faeces samples, previously tested using a golden standard ELISA, the Premier™ Rotaclone^®^ (Meridian Bioscience™, Inc., Cincinnati, OH, USA), and by PCR assay. Faecal suspensions (10% *w/v* in 100 mM Tris, 1.5 mM CaCl_2_; pH 7.2) were centrifuged at 5000× *g* for 20 min at 4 °C, and the supernatants (25 µL) were added to 25 µL of the CLB-bound anti-RVA IgY into a latex agglutination plaque. The mixture was carefully homogenized for 5 min, incubated for 30 min, and observed for the presence or absence of agglutination by comparison with negative control, positive control (RVA-positive faeces sample), and the latex control (normal goat serum conjugated with latex + faecal suspension). All samples were assayed in duplicate. The Oswaldo Cruz Foundation Ethical Committee (CEP-FIOCRUZ) approved the present study (number 311/06).

## 3. Results

### 3.1. Clinical Monitoring and Immunisation Effects on Hen’s Biology

All immunised hens were healthy throughout the experiment, without pain or discomfort at palpation, nor tissue damage in the inoculation area. Animals inoculated with rotavirus showed no difference in growth kinetics compared with those receiving only the non-immunogenic inoculum (control group). The histological analysis showed congestion and diffuse steatosis in the liver parenchyma of both the hens that received immunogen (RVA) and those that did not. The kidney samples did not show changes, whereas the spleen showed congestion.

### 3.2. Characterisation of the Purified Anti-Rotavirus IgY

As the egg-laying period began (at 138 days of life, 35 days after the last immunisation), 477 eggs were collected throughout 13 weeks. After purification of the yolk protein by precipitation with PEG 6000, the total protein concentration was measured per group. A peak of protein concentration was observed in the ninth week and remained high until the last week. Group I, which received three immunisation doses with the immunogen (RVA), had a higher total protein concentration than the other groups throughout the egg collection, and the highest total protein concentration (59.5 mg/mL) in the 10th week. There were significant differences in protein concentration means among the three groups (*p* < 0.05): 37.35 mg/mL (Group I), 31.58 mg/mL (Group II), and 27.46 mg/mL (Group III). 

The proteins purified from egg yolks reproduced the characteristic patterns of the birds’ IgY SDS-PAGE electrophoretic patterns, regardless of the group (I, II, or III) of hens. Figure 1 depicts the electrophoretic pattern of the IgY, showing light and heavy chain molecular weights of 25 kDa and 70 kDa, respectively [20]. Although low amounts of other proteins had been observed, the IgY chain bands were denser.

The specificity of anti-RVA IgY purified from egg yolks was confirmed by Western blotting (Figure 2). Both groups I and II, which received rotavirus as an immunogen, had anti-RVA binding to human and simian rotavirus proteins VP1 (125 kDa) and VP6 (44.8 kDa) [5], better evidenced by antigen–antibody staining. In contrast, no anti-RVA IgY binding was evidenced for the control group.

In order to assess the anti-RVA IgY specificity and efficacy in blocking RVA entry in MA-104 cells, and to obtain the ideal concentration of IgY required to neutralize the virus, an in vitro neutralization assay was performed. RVA replication was detected at IgY titres of 1.5 mg/mL (1.94 × 10^3^ RVA copies/mL), 1.0 mg/mL (6.61 × 10^3^ RVA copies/mL), 0.75 mg/mL (5.19 × 10^3^ RVA copies/mL), 0.5 mg/mL (3.24 × 10^3^ RVA copies/mL), and 5.05 × 10^5^ RVA copies/mL) of the positive control. Hence (above 1.5 mg/mL of IgY neutralizing antibody titre), RVA replication was not detected.

We designed an in-house latex agglutination assay using the purified anti-RVA IgY bound to carboxyl latex beads, which showed concordant results (absence or presence of agglutination) against the negative, positive, and agglutination control (normal goat serum conjugated with latex + faecal suspension), respectively (Figure 3). Compared with the Premier™ Rotaclone^®^ ELISA for antigen detection in human faecal samples, our anti-RVA IgY latex beads performed concordant results with 75% (3/4) ELISA and PCR-positive and 87.5% (14/16) negative samples.

## 4. Discussion

Egg yolk immunoglobulin (IgY) has many advantages over producing monoclonal and polyclonal antibodies from mice and rabbits, respectively, as a supply for diagnostic and immunotherapy purposes. Considering the 3R principle of Replacement, Reduction, and Refinement, IgY production represents an ethical alternative to substitute mammals [21]. As reported in the literature, the benefits of IgY technology for polyclonal antibody production and application in diagnostic and passive immunotherapy motivated the present study. 

Human (RVA/human-tc/USA/Wa/1974/G1P [8]) and simian (RVA/Simian-tc/ZAF/SA11/1958/G3P [2]) rotaviruses were propagated in MA-104 cell culture on a large scale, and the caesium chloride concentration method refined the virus purity degree [15]. The absence of cellular debris and proteins from cell culture was essential for obtaining the viral inoculum. Due to the phylogenetic distance, conserved mammalian proteins exhibit high immunogenicity in birds [22], since such interfering antigens would promote a robust immune response. They are not, however, entirely effective against rotavirus.

In this study, hens received their first immunisation at 38 days old, when birds are less susceptible to stress by handling, thus avoiding a decrease in egg-laying [23]. Being aware that the bursa of Fabricius and adaptive immune system are immature before 8–10 weeks of age [24], we took the risk. Nevertheless, the antibody yield and the immunoglobulin avidity upgraded after the second (at 74 days old) and third (at 103 days old) immunogenic doses. Regarding the IgY immunoglobulin concentration, the results achieved in our study corroborate those described by Schade and colleagues, in which the antibody titre significantly increased after a booster immunisation [2]. In an earlier study, we achieved a high yield of hepatitis A virus-specific IgY following the same immunisation schedule [14]. It is well accepted that multiple immunisations might be needed even for the most successful vaccines. Over the past decade, studies have shown that prime–boost immunisations can be performed with different vaccines containing the same antigens in a “heterologous” prime–boost format [25,26,27]. Interestingly, most of the time, heterologous prime–boost is more effective and more immunogenic than the “homologous” prime–boost approach [28]. However, the heterologous effectivity was not confirmed in SARS-CoV-2 [29].

Immunisation of the hens occurred without clinical manifestation and within a regular kinetics growth line. The egg-laying period occurred, as expected, at 138 days of age. The histological finding of steatosis and liver congestion in the immunised and control groups are questionable. Most likely, such non-specific histological findings might be associated with the commercially available poultry feed, which is rich in fat [30], not due to inoculations. Hence, we assume that the rotavirus inoculation did not induce liver injury. 

The egg-laying began thirty-five days after the last immunisation. The protein concentration peaked at the 9th and 10th weeks in all groups, and the concentration decreased up to the 13th week. These results corroborate a previous study [31]. Hens that received three immunogenic doses showed a protein concentration average (37.35 mg/mL) higher than those receiving two immunogenic doses and the non-immunogenic inoculum. It is important to emphasise that we used a precipitation-based method for the egg yolks. Thus, proteins other than the immunoglobulin IgY were purified. Nevertheless, although the vast majority of the proteins purified are IgY, as confirmed by electrophoresis, RVA-specific immunoglobulin Y (IgY) represents 2 to 10% of the amount of purified IgY [32].

The presence of IgY purified from the egg yolk was assessed by SDS-PAGE according to the molecular weight profile of their light and heavy chains. The IgY light chain has a molecular weight of 25 kDa and the heavy chain 70 kDa [20]. Other proteins were observed in lower quantities. It is important to reinforce that the proteins purified from egg yolks were 1:100 diluted in SDS-PAGE. Due to this, the electrophoretic bands corresponding to light and heavy chains were thicker than those corresponding to other proteins. Thus, in this study, IgY was the main protein purified from the egg yolk, whereas previous studies had described proteins other than IgY co-purified by PEG [33,34]. The IgY yield is not the unique significant aspect of production; part of the IgY must be specific to the rotavirus antigens, because the deposition of any IgY in the egg yolk occurs naturally in birds in order to confer protection to the embryo. The specificity of the purified IgY to both human and simian rotaviruses VP1-VP7 was assessed by Western blotting. As depicted in Figure 2, the IgY bound specifically to viral proteins, largely VP6 [5], the most immunogenic and antigenic rotavirus protein [35,36].

Aiming to promote appropriate immune responses at both innate and adaptive levels, we combined the incomplete Freund′s adjuvant (IFA) with the synthetic oligodeoxynucleotide (CpG-ODN), a synthetic agonist of Toll-like receptor 9, which was shown to be efficient in B-cell stimulation and humoral response. It is also a potent inducer of innate and acquired immunity [37,38]. It has been shown that CpG-ODN may function as an effective adjuvant for vaccines against a variety of pathogens, such as bacteria, viruses, fungi and parasites [39,40,41,42,43]. Both adjuvants not only induce the maturation of antigen presentation cells but also trigger the differentiation of Th0 to Th1. Thus, both the humoral and Th1 responses may be enhanced [44]. In a previous study, we described a higher production of anti-HAV IgY in birds immunised with CpG-ODN [14].

Regardless of the inoculum formulation used, this experiment provided a consistently high yield of antibodies, with each group laying approximately 180 eggs throughout 13 weeks. Considering that the egg yolk contains 100 to 150 mg of IgY antibodies, such an egg production can result in 24 to 36 g of IgY per year per hen [45,46]. However, the specific IgY portion represents 2 to 10% of the total IgY obtained [32]. In our experiment, the yield of specific IgY increased onward in the experiment so that the antibody titre remained high, even after a long time without immunisation. In addition, the neutralisation ability was confirmed, since the purified IgY blocked the functional activity of RVA in the MA-104 cell lineage. The neutralization occurred up to the concentration of 2.0 mg/mL of total proteins, whose largest amount might have been corresponding to IgY; at 1.5 mg/mL, the viral load remaining after neutralisation was very low (1.94 × 10^3^ copies RNA/mL). Dai and colleagues found a comparable result with a rotavirus-specific IgY, with neutralization occurring at 1.0 mg/mL [47,48]. 

There are few studies in the literature regarding the use of IgY in latex agglutination tests [49,50,51]. In one study, authors assume that the substitution of mammalian antibodies (IgG) by avian antibody (IgY) improved the latex-agglutination test, and the enzyme immunoassay (EIA) to C-reactive protein (CRP), with fewer false-positive results [52]. The absence of the rheumatoid factor when comparing a latex test using chicken IgY with IgG from several mammalian species could explain this feature [53]. We evaluated the ability of the anti-RVA IgY resulting from our experiment to detect rotavirus antigens using a preliminary latex agglutination-based assay. In due course, and as we aim to apply our anti-RVA IgY to latex agglutination or enzyme-based immunoassay format, its sensitivity, specificity, and reproducibility must be evaluated against a larger panel of well-characterised samples.

Rotavirus-induced diarrhoea is a major public health concern worldwide. There is cumulative evidence toward the oral administration of IgY as a supportive treatment of paediatric diarrheal diseases [54]. There is, however, a clear need for clinical trials to confirm the anti-rotavirus IgY efficacy and safety as passive immunotherapy. To assess the therapeutic efficacy of the purified anti-RVA-IgY, we conducted an in vivo experiment using young cynomolgus monkeys treated with one or two doses of this IgY and challenged with human RVA. Preliminary results showed the absence of episodes of diarrhoea, which is the gold standard for clinical efficacy. Virological and histopathological findings are under analysis.

## Figures and Tables

**Figure 1 viruses-14-01995-f001:**
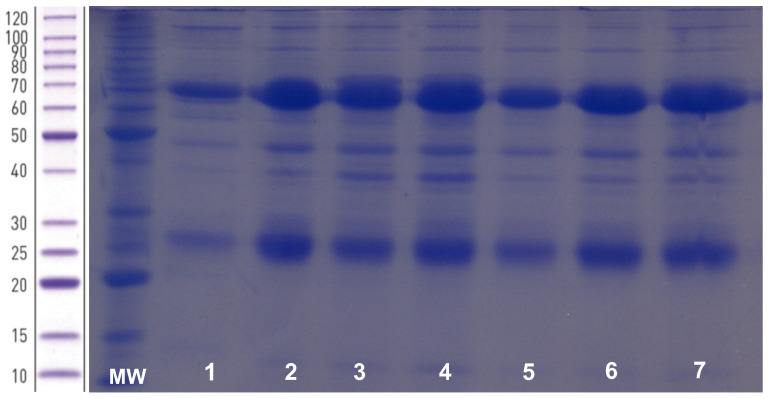
SDS-PAGE electrophoretic pattern of purified egg yolk proteins. MW: molecular weight standard (BenchMark™ Protein Ladder-Invitrogen^®^, Carlsbad, CA, USA). Lanes 1–7: IgY Group I weeks 1–7 (diluted 1:100).

**Figure 2 viruses-14-01995-f002:**
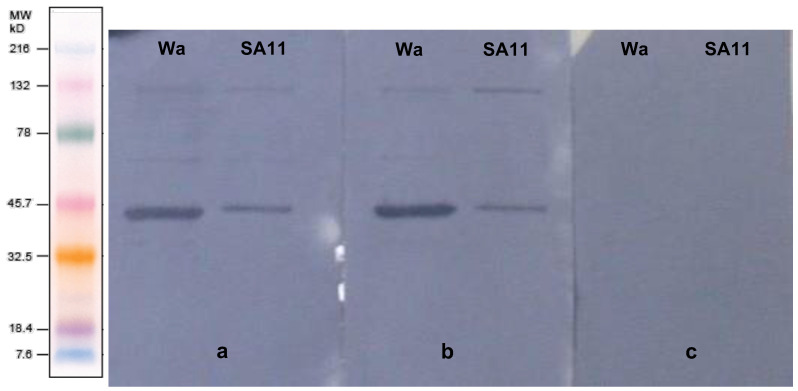
Western blot of anti-RVA IgY. Wa and SA11: human and simian rotavirus proteins stained with specific IgY. (**a**) Group I (three RVA doses); (**b**) Group II (two RVA doses); (**c**) Group III (negative control); MW: molecular weight standard (Kaleidoscope™ Pre-stained Standards—Bio-Rad Laboratories, Hercules, CA, USA).

**Figure 3 viruses-14-01995-f003:**
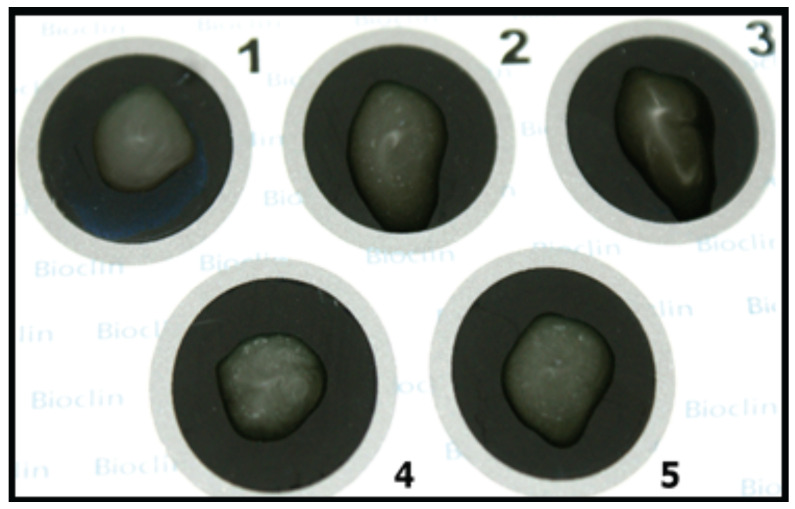
Anti-RVA IgY latex bead agglutination assay: (**1**) negative control; (**2**) rotavirus-positive control; (**3**) agglutination control (normal goat serum conjugated with latex + faecal suspension); (**4**,**5**) RVA-positive faecal samples.

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
