# Peer review of "Production and Evaluation of Chicken Egg Yolk Immunoglobulin (IgY) against Human and Simian Rotaviruses"

_viruses, 2022, doi:10.3390/v14091995_

Round 1

Reviewer 1 Report (Previous Reviewer 1)

There are some typographical errors which should be corrected.

Line 65. 1010-1011 should be 1010-1011.

Line 147. “0,5-0,75-1,0-1,5-2,0-2,5-3,0-3,5-4,0” should be  “0.5-0.75-1.0-1.5-2.0-2.5-3,.0-3.5-4.0”.

Lines 165 and 179. The word, ” reaction, is not adequate.

Line 54. The diameter of rotavirus particle should be 100 nm in diameter.

Author Response

Reviewer 2 Report (New Reviewer)

The manuscript describes de immunization of hens to obtain human and simian rotavirus-specific IgY capably of neutralizing virus infection in vitro. The article is well written but there are some important considerations to take into account. – 1. Even though the authors use a different immunization strategy, RVA-specific IgY production was clearly described before (Vega CG, Bok M, Vlasova AN, Chattha KS, Fernández FM, Wigdorovitz A, Parreño VG, Saif LJ. IgY antibodies protect against human Rotavirus induce diarrhea in the neonatal gnotobiotic piglet disease model. PLoS One. 2012;7(8):e42788. doi: 10.1371/journal.pone.0042788. Epub 2012 Aug 3. PMID: 22880110; PMCID: PMC3411843.; "Evaluation of hyperimmune Hen Egg Yolk Derived Anti-Human Rotavirus Antibodies (Anti-HRVIgY) against Rotavirus Infection" written by Manika Buragohain, Ganesh S. Dhale, Govind R. Ghalsasi, Shobha D. Chitambar, published by World Journal of Vaccines, Vol.2 No.2, 2012).

2. 2 inoculum and immunization: it is not clear how much virus was used in each dose. It says for example 2.2 x10E5 copies, is this per ml? per dose? Was it measured by qPCR?

2. 3. IgY characterization: authors measure total proteins but not RVA-specific IgY. There is not an assay to evaluate the hens` immune response to de antigens.

When the authors explain in vitro VN assay they mention dilution of anti-rotavirus IgY (and it is mention doubly) when they never measure it specifically. The only quantification is total protein.

3.1 Clinical monitoring: it is mentioned that all animals showed liver and spleen histological modifications. Authors conclude that steatosis is due to feeding but they never mention evaluation of tissues from unvaccinated animals, also, they don’t show pictures of different tissues.

3.2 Characterization of the IgY:

Line 172à mean concentration of what? Total protein? Why do the authors think the total protein should be different among groups? How were statistics calculated? Authors show average but not SD.

Figure 1: It is not shown other groups’ samples

There is no figure showing RVA-specific IgY hen’s response. This is a key result to show.

Figure 2: There is less VP6 RVA-specific IgY against SA11 strain than Wa strain. Is there any explanation for that being the same amount of virus in each vaccine?

Line 194 à Results should be re-written. It mentions IgY titers when authors only measure total protein. The authors don’t mention the “IgY neutralizing concentration” in the results section (1.5 mg/ml is not a titer). They mention up to 2 mg/ml in the discussion section. There is no graph or table showing these results. The authors don’t mention the control group results.

Latex Agglutination: “..showed consistent results” with what? Samples tested with which other test? The authors don’t show a detailed result and Figure 3 is not easy to read. The concordance of the technique should be analyzed using statistic software and validation parameters should be detailed.

4. Discussion: It mentions the cesium chloride concentration of RVA which is not mentioned in the M&M section.

HAVà define

The authors don’t discuss hens’ immune response to the antigens.

Line 253 à explain. When IgY is diluted also co-purified proteins are diluted.

Line 257 à how much IgY is specific to RVA?

CpG-ODN: Authors discuss using this adjuvant without showing any result in RVA-specific immune response.

Line 270 à “consistently high yield of antibodies” the authors didn’t measure the amount of IgY

Line 274 à the statement “the yield of specific IgY increased…” is wrong because IgY wasn’t measured

Authors keep talking about IgY concentration when only total protein was measured.

Line 280: the authors should consider to add the following citation (Vega CG, Bok M, Vlasova AN, Chattha KS, Fernández FM, Wigdorovitz A, Parreño VG, Saif LJ. IgY antibodies protect against human Rotavirus induce diarrhea in the neonatal gnotobiotic piglet disease model. PLoS One. 2012;7(8):e42788. doi: 10.1371/journal.pone.0042788. Epub 2012 Aug 3. PMID: 22880110; PMCID: PMC3411843)

CRPà define

Round 2

Reviewer 2 Report (New Reviewer)

The manuscript shows the development of IgY against human and simian rotavirus. The authors can explain each comment made by this reviewer in the first revision, but the answers are not well reflected in the manuscript.

I understand that some answers are in the bibliography cited but it would be nice to have some of those details within the manuscript (eg virus/dose or virus purification).

The authors dont measure RVA-specific IgY response. It is a very easy assay to perform (ELISA), and even when the authors explain that specific IgY is being measured by VN assay they don't discuss a VN titer. There is no mention in the discussion section that 2-10% of that total protein measure is supposed to be RVA specific antibody. 

It is well known that IgY antibodies to RVA can neutralize the infection. I Think that authors should add some new information to IgY characterization or give a different insight to have a more interesting publication.

Author Response

Response to Reviewer 2 Comments

Point 1: The manuscript shows the development of IgY against human and simian rotavirus. The authors can explain each comment made by this reviewer in the first revision, but the answers are not well reflected in the manuscript.

Response 1:
Regarding the clinical monitoring (histological changes): We added comments about the histological changes, in results (lines 167-170) and discussion (lines 252-256).

Results section, Characterization of the purified anti-rotavirus IgY: The proteins purified from egg yolks reproduced the same SDS-PAGE electrophoretic patterns characteristic of birds IgY regardless the group (I, II, or III) of hens. Text was rewritten accordingly (lines 181-187).
Regarding the IgY electrophoretic pattern after dilution: the text was rewritten accordingly (lines 264-265): “Hence, not only the immunoglobulin IgY but other proteins got purified. Nevertheless, although the vast majority of the proteins purified are IgY, confirmed by electrophoresis, RVAspecific immunoglobulin Y (IgY ) represents 2 to 10% of the whole amount of the purified IgY [32]”.

Regarding the use of CpG-ODN: We have succesfully used this adjuvant in hens’ immunisation with the hepatitis A virus (HAV). In this previous study, we compared groups of hens immunized with and without CpG-ODN, and observed that the highest amount of specific IgY was obtained in hens which were immunized with this adjuvant (de Paula, V.S.; da Silva, A. dos S.; de Vasconcelos, G.A.L.B.M.; Iff, E.T.; Silva, M.E.M.; Kappel, L.A.; Cruz, P.B.; Pinto, M.A. Applied Biotechnology for Production of Immunoglobulin Y Specific to Hepatitis A Virus. J. Virol. Methods 2011, 171, 102–106, doi:10.1016/j.jviromet.2010.10.008). According to the referee advice, we cited our paper (reference 14: De Paula et al., 2011) in the Discussion section (lines 286-287). Actually, the advantage of using the CpGODN had already been discussed briefly in the current article. 

Regarding the Referee’s advice to detailing the “validation parameters” of the latex agglutination assay (preliminarily performed): We clarify that, in the present study, we designed a preliminary inhouse latex agglutination-based assay with the only purpose of assessing the ability of the anti-RVA IgY to react against RVA-positive samples specifically, but not with negative samples. Although we had informed in our manuscript that the results are preliminary, we agree that a larger sample panel would have been necessary for validation. However, it was not our purpose in the present study. The text was rewritten in the Result and Discussion sections (lines 209-212 and 304-310, respectively).

Point 2: I understand that some answers are in the bibliography cited but it would be nice to have some of those details within the manuscript (eg virus/dose or virus purification).

Response 2: We added detailed informations, as suggested by the referee (lines 87-90): “ …inoculum were produced in cell culture using the monkey African green kidney lineage (MA-104-ATCC® CRL2378) and purified with caesium chloride gradient by ultracentrifugation at 240,000 x g, as described previously [15]. The final concentrations of Wa RVA and SA11 RVA solutions were,  respectively: 2.2x1010 and 2.0x108 RNA copies/ml”.

Point 3: The authors dont measure RVA-specific IgY response. It is a very easy assay to perform (ELISA), and even when the authors explain that specific IgY is being measured by VN assay they don't discuss a VN titer. There is no mention in the discussion section that 2-10% of that total protein measure is supposed to be RVA specific antibody.

Response 3: We would have measured the anti-RVA IgY response by other methods than in vitro neutralization. However, our bigger goal was to evaluate the safety and efficacy of our IgY as alternative profilaxy and treatment of a non-human-primate (NHP) model experimentally infected with rotavirus. Since there was not enough amount of IgY after our NHP experiment, and since the in vitro neutralization assay succeeded, such an estimation by, e.g., ELISA had to be postponed.

Regarding the referee´s argument “There is no mention in the discussion section that 2-10% of that total protein measure is supposed to be RVA specific antibody”, it might have been misunderstanding. Indeed, we have mentioned twice in the Discussion section that: “… not only the immunoglobulin IgY but other proteins got purified. But, the vast majority of the proteins purified are IgY, confirmed by electrophoresis, and RVA-specific IgY represents 2 to 10% of total IgY”. To make it clear, the text was rewritten as: “Hence, not only the immunoglobulin IgY but other proteins got purified. Nevertheless, although the vast majority of the proteins purified are IgY, confirmed by electrophoresis, RVA-specific immunoglobulin Y (IgY) represents 2 to 10% of the whole amount of the purified IgY” (lines 262-265). Once again, in the Discussion section, we had mention that “However, the specific IgY portion represents 2 to 10% of the total IgY obtained [32].” (lines 291- 292). (…) “The neutralization occurred up to the concentration of 2.0 mg/ml of IgY;”. To make it clear, the text was rewritten as: “The neutralization occurred up to the concentration of 2.0 mg/ml of total
proteins, whose largest amount might have been corresponding to IgY;…” (lines 295-296).

Note: The assumption that ”approximately 2-10% of the total protein quantified is specific IgY” is according to: Schade, R.; Bürger, W.; Schöneberg, T.; Schniering, A.; Schwarzkopf, C.; Hlinak, A.; Kobilke, H. [Avian Egg Yolk Antibodies. The Egg Laying Capacity of Hens Following Immunisation with Antigens of Different Kind and Origin and the Efficiency of Egg Yolk Antibodies in Comparison to Mammalian Antibodies]. ALTEX 1994, 11, 75–84)”.

Point 4: It is well known that IgY antibodies to RVA can neutralize the infection. I Think that authors should add some new information to IgY characterization or give a different insight to have a more interesting publication.

Response 4: Indeed, we assessed the RVA-specific IgY ability to neutralize rotavirus infection using NHP (cynomolgus monkeys) immunised with our IgY. As mentioned in the final paragraph of our manuscript: “To assess the therapeutic efficacy of the purified anti-RVA-IgY, we conducted an in vivo experiment using young cynomolgus monkeys treated with one or two doses of this IgY and challenged with human RVA. Preliminary results showed the absence of episodes of diarrhoea, which is the gold standard for clinical efficacy. Virological and histopathological findings are under analysis”.

This manuscript is a resubmission of an earlier submission. The following is a list of the peer review reports and author responses from that submission.

Round 1

Reviewer 1 Report

This manuscript describes the application of IgY from hens immunized with rotavirus for latex agglutination assay. The authors described some merits of the use of IgY for the latex agglutination assay for rotavirus detection.

The manuscript was written well. However, as the authors described, latex agglutination assay for the detection of rotavirus was developed in 1981. In addition, the utility of the latex agglutination assay for rotavirus is not high, since various convenient assays for detecting rotaviruses have been developed.

The number of the stool specimens from patients positive for rotavirus is too small for examining the specificity and sensitivity of the latex agglutination assays using IgY. At least 20 rotavirus-positive samples are necessary for the examination for that purpose. In addition, the analysis of the results showing discrepancy with the results in a gold standard assay should be performed.

The manuscript is lengthy. For example, Table 1 is not necessary, since the protein concentrations in eggs are not informative. References are too many. The authors can shorten the manuscript in various sections. 

Abstract, lines 31-33. Page 7, lines 264-267. Is the sentence adequate? The latex agglutination assay detected 14 of 16 RV-negative samples.

Figure 4. Page 7, line 273. What does the expression “faecal panel samples (challenge)” in the legend mean?

Reviewer 2 Report

Dear authors,

 At present, avian IgY is widely commercilized for anti-viral sustances and anti-parasite therapy,including preparation,purification and powder formations with long time storage. The present IgY anti-rotavirus for bead agglutination assay is a simple test for antigen diagnosis. However,some missing data are hampering the submission quality and acceptance. I have to draw your attentions to the following comments and suggestions.

1.IgY trend post immunization. The reviewer did not see the IgY levels post immunization with two different boosting strategies. Please provide IgY levels using ELISA kit. The protein concentrations in Table 1 are inconsisted with IgY levels.

2.As for specific test of bead agglutination assay, other gastroenteritis-inducing infectious agents are required to be distinguised, such as noroviruses, enteric adenoviruses 40 and 41, astroviruses, Escherichia coli, Salmonella spp and Bacillus cereus. Moreover, cross-reaction among 10 different rotavirus species(A-J) must be done to confirm the specifity test.

3.Clinical trial in comparision with commercial ELISA or PCA method. The authors have to do clinical test compared to standard ELISA kit or PCR assay.